# The Rationale for the Optimal Continuous-Variable Quantum Key Distribution Protocol

Roman Goncharov *, Irina Vorontsova, Daniil Kirichenko, Ilya Filipov, Iurii Adam, Vladimir Chistiakov, Semyon Smirnov, Boris Nasedkin, Boris Pervushin, Daria Kargina, Eduard Samsonov and Vladimir Egorov

Leading Research Center "National Center for Quantum Internet", ITMO University, Kronverkskiy 49, St. Petersburg 197101, Russia
* Correspondence: rkgoncharov@itmo.ru

**Abstract:** This article describes the current technical level of developments in the field of continuous-variable quantum key distribution (CV-QKD). Various classifications are described, the criteria are analyzed, and the optimal protocol is selected. The analysis is focused around device-dependent schemes with a theoretical emphasis, and therefore, a detailed analysis of device-independent CV-QKD and side-channel attacks is out of the scope of the work. However, the latter, one way or another, is taken into account when describing possible classifications. The choice of the optimal protocol was carried out, first of all, from the potential possibility of integration into existing network telecommunication infrastructures. Predominantly, the general classification is carried out in such a way that it is possible to draw up a specific protocol, depending on the task of implementation.

**Keywords:** quantum key distribution; continuous variables; security proofs

## 1. Introduction

Currently, quantum communications, in general, and quantum key distribution (QKD) as one of the internal directions, in particular, are some of the most actively developing areas of quantum technologies [1,2]. QKD allows one to send a secure key between several legitimate users connected by so-called quantum and classical channels. Theoretically, the security of QKD is based on the principles of quantum mechanics, which guarantees security against any unforeseen technological developments, for example, in the field of quantum computing [3]. Although the development of QKD technologies has not reached its peak yet, many different solutions are already being offered, starting from conventional fiber-optic quantum communication systems using the well-known BB84 protocol [4–6], and ending with complex distributed quantum networks [7] and other systems built in a non-trivial way [8–10]. That is, the scope of QKD applications is expanding every year, as well as the number of approaches to its implementation. This is dictated by the need to find affordable alternative approaches to information protection, in which validity should be comparable to the classical solutions that have become canonical in the modern world. Undoubtedly, QKD systems will not replace the existing infrastructure, but be integrated into it. In this regard, an impressive proportion of current and planned potential research in the field of quantum communications is devoted to finding solutions to many practically significant problems. One example is the use of coherent detection methods in QKD systems, implemented by devices used in classical fiber-optical communication systems, instead of single-photon detectors, that are technically complex and expensive devices. Such QKD systems based on a coherent detection method [11–17] are called continuous-variable QKD (CV-QKD) systems, while systems that rely on a single photon detection are called discrete-variable QKD (DV-QKD). Coherent detection itself seems to be a well-studied method, where the signal, thanks to simple manipulations with the beam splitter, is amplified proportionally to the coherent reference signal—the local oscillator (LO) [18].

Protocols that imply single-photon detection will be referred to as discrete protocols or QKD protocols on discrete variables (DV-QKD). Of note, CV-QKD systems can be implemented in the concept of measurement-devise-independent (MDI) protocols [19–22]. The idea behind such an approach is an opportunity for the parties to recover the information about their variables based on the parameter, which is obtained after the procedures of a Bell-type measurement (i.e., homodyning of the quadratures of Alice and Bob's modes after they passed through a beam splitter) at an untrusted relay followed by a measurement of the displacement operator. This parameter relates the coherent states of Alice and Bob to to locally rebuild the sender–receiver covariance matrix without any information disclosure. However, CV MDI QKD will not be considered further in the work, for such solutions are inconvenient practically and feasible economically. However, we have considered this type of protocol separately in a related paper [23].

There is a number of obvious advantages of the CV-QKD [2] approach related to the planned scientific and technical level of development, including cost reduction, high secure key generation rate, and scalability.

This article is devoted to the analysis of the current state of development in the field of device-dependent CV-QKD systems, and the results achieved so far to identify the potential implementation into existing network telecommunication infrastructures. In Section 2, the criteria for choosing the optimal CV-QKD protocol are formulated and justified, each of which is given a separate description containing overview and analytical information. In conclusion, the optimal conditions satisfying the claimed and discussed criteria of the CV-QKD protocols are summarized.

## 2. Classification of CV-QKD Protocols and Justification of Optimal Parameters

Regarding DV-QKD, CV-QKD protocols can be classified according to various parameters. Main criteria to classify CV protocols are presented in Table 1 and will gradually be disclosed further. Each classification criterion presented in Table 1 is assigned a separate subsection. At this stage, it should be noted that the protocols can be equivalent from a mathematical point of view; that is, the same protocol can be implemented in two different ways but their description can be reduced to a single one (a similar trend is observed in the case of DV-QKD protocols).

**Table 1.** Possible classifications of CV-QKD protocols.

| Classification | Item |
| --- | --- |
| general scheme of a protocol | prepare-and-measure; entanglement-based |
| quantum channel implementation | fiber-optical; free space |
| channel configuration | one-way scheme; two-way scheme |
| type of modulation | Gaussian; non-Gaussian |
| signal states | single-mode squeezed; single-mode coherent; multimode coherent; two-mode squeezed; thermal |
| coherent detection schemes | homodyne; heterodyne (double homodyne); heterodyne at intermediate frequency |
| LO implementation | on Alice's side; on Bob's side |
| reconciliation protocols | direct (or forward); reverse |

### 2.1. General Approach to CV-QKD Protocol Description

As for a general classification of QKD protocols, CV-QKD included, two scenarios can be distinguished: "prepare-and-measure" (PM) and "entanglement-based" (EB) ones. In the PM scenario, Alice encodes classical information into quantum states (the preparation stage), then transmits them to Bob as optical signals. Subsequently, Bob makes a series of measurements for each of the received signals to restore the encoded information. EB scenario, in turn, implies that both Alice and Bob operate with modes of the entangled

state. CV-QKD being the case, Alice generates a two-mode squeezed vacuum state and measures both quadratures of one of its modes. In turn, the second mode is sent to Bob to be projected onto a coherent state. In practice, most CV-QKD systems belong to the first of the above-mentioned approaches. Nevertheless, both for DV- and CV-QKD protocols, it is convenient to use the virtual entanglement method. The interchangeability of EB and PM approaches for the CV-QKD case is proved in the paper [24] for single-mode coherent states in the PM scheme and distribution of two-mode squeezed vacuum states in the EB scheme. The covariance matrices describing the two above-mentioned scenarios coincide up to a constant.

A similar relationship exists for DV-QKD protocols as well. For example, BB84 protocol can be reduced to the equivalent EB version, where Alice and Bob operate with an entangled pair of photons [25]. Additionally, it is possible to use entropic uncertainty relations uniting information leakage to Eve and classical conditional entropy between Alice and Bob. This approach is equivalent to a protocol where Alice randomly and equiprobably selects a basis, and then selects one of three states within the basis, also randomly and equiprobably, and sends them to the channel. The possibility of using entropic uncertainty relations for three-particle systems describing the general quantum state "Alice–Bob–Eve" exists only by reducing the original PM protocol to a mathematically equivalent EB version [26].

The equivalence discussed cannot be called completely strict when considering a broader class of attacks [27], which implies the need for a security proof in the PM scenario [28–30]. However, security against collective attacks for finite keys has been proved for CV-QKD protocols both in PM and EB approaches separately [31–33].

If we address the protocol on squeezed states with the homodyne detection method [34], although the security proof against coherent attacks for finite-length keys was demonstrated, the secure key generation rate assessment turned out to be too pessimistic; that is, $\lim_{N \to \infty} K^{\varepsilon}(N) < K_{\text{call}}^{\text{asymp}}$.

The recently proposed de Finetti Gaussian reduction approach for CV-QKD protocols with Gaussian modulation [31] shows the possibility of achieving security against coherent attacks in realistic implementation using the protocol invariance considering a unitary group instead of a symmetric one. That is, the protocol part has been simplified in the terms of symmetrization. A mention should also be made of additional energy tests (measurements of the local number of photons on the users' sides) in the context of the security described. Moreover, recent results show that certain parameters can provide a key distribution with a sufficient key rate, taking into account the finite key effects, confirmed by numerical modeling methods [27,35–37]. For real practical systems, a composable security is provided against collective attacks (without symmetrization and energy tests) by the current moment [17].

Noteworthy, here, an important motivation to account for in terms of a protocol choice is the need for a convenient and as "seamless" as possible integration of the CV-QKD system with existing telecommunication infrastructure. This implies the search for solutions that are simpler in the context of state generation (namely, single-mode coherent states), rather than two-mode squeezing. The latter option is, indeed, a frontier solution of great interest; however, the complexity of such states' preparation forces us to reject this approach. Thus, the choice is made in favor of the PM scheme for the above-discussed mathematical equivalence to EB schemes.

### 2.2. Quantum Channel Implementation

There are two ways to implement a quantum channel for CV-QKD, as well as for DV-QKD: through optical fibers [17,38] and optical channels in free space [27,39]. The latter is attracting more and more attention because of the convenience of creating infrastructure in practice; unlike fiber channels, transmission through free space is devoid of all the disadvantages associated with transportation and difficulties in installing a physical fiber channel sensitive to mechanical influences and seismic features of the terrain (here, underground lines). This, undoubtedly, makes them more portable and flexible in the context

of installation and operation. However, when working with free space, it is necessary to take into account not only the presence of diffraction losses, atmospheric extinction, and background thermal noise [40,41], but also the attenuation effect caused by guidance error and turbulence [42].

Fiber systems for secure key transmission, in turn, have already proven themselves in the context of QKD: such systems provide a high level of durability (the vulnerability of fiber-optic QKD communication systems is analyzed in their study), low cost of implementation, and availability of installation and maintenance.

Additionally, though free space realizations of CV-QKD protocols carry a huge research potential and have their own advantages, we do not consider them as the main option here, for we aim at the integration of CV-QKD systems with existing telecommunication fiber optical networks.

### 2.3. Channel Configuration Schemes

For the case of DV-QKD, two schemes can be implemented: one-way and two-way schemes. Moreover, both schemes can implement the same protocol; for example, there are one-way schemes for BB84 protocols (for example, Ref. [43]) and two-way schemes using a QKD system of plug-and-play [6,44] type.

Similar schemes can be implemented in the CV-QKD case as well. In a typical one-way CV-QKD protocol, Alice prepares a quantum state and sends it to Bob via a quantum channel. The receiver performs coherent detection, followed by post-processing procedures performed by users.

One disadvantage of one-way schemes is the presence of phase and polarization distortions in the channel. To mitigate them, it is necessary that additional algorithms and modifications be used. Despite the fact there are questions about the stability of the system, for one-way CV-QKD schemes with Gaussian modulation, security against coherent attacks is justified [2,31].

In the two-way CV-QKD scheme [45–47], analogously to DV-QKD ones, Alice and Bob use a quantum channel twice to obtain a raw key. Initially, Bob prepares the so-called reference states and sends them to Alice through a quantum channel. Alice, in turn, encodes the information by applying unitary operations to the received reference states, and then sends them to Bob for measurement. In such CV-QKD schemes, various information encoding protocols can be used; the common ones are Gaussian modulation of coherent states, implying the use of amplitude and phase modulators, as well as double phase modulation, implying two phase modulators [44,48].

The main advantage of the two-way scheme is the potential stability of the QKD system, which is achieved by the auto-compensation of phase and polarization distortions of light passing through the same fiber in two directions. However, in the framework of world scientific practice, such QKD systems are only nascent [48,49], and have not yet been sufficiently studied. Moreover, two-way schemes can be limited in the implementation of a LO (preference is given to local LO, see Section 2.7) since the transmitted LO, like the signal, suffers twice as much losses in two passes. A practical security proof of two-way CV-QKD systems is also an actual task.

### 2.4. Types of Modulation

In the context of choosing the type of modulation of the CV-QKD system, it can be conditionally divided into two actively developing areas [50]: CV-QKD protocols with Gaussian [11,13–17,51] and discrete modulation [12,52–59].

At the same time, a critical disadvantage of this type of CV-QKD protocol is the incompleteness of their theoretical models. However, to date, active work are underway to fill this gap. There are a number of works demonstrating different approaches to proving security in the asymptotic regime [53,54,60] (i.e., for keys of infinite length) against an optimal attack (the convex optimization problem is solved). It is important to note that these works have already provided the basis for the composable security proof, including

non-perfect homodyning and finite-key effects in the context of collective attacks [57]. A security proof using de Finetti for discrete modulation is still an open problem.

In addition, discrete modulation is inferior to the Gaussian one in terms of the secure key generation rate and the maximum distance at which a key distribution session is still possible: the results are demonstrated for various configurations of QAM and PSK (see Ref. [54]).

The unconditional security of CV-QKD protocols with Gaussian modulation, on the contrary, is strictly proved taking into account the finite key effects. Most recent experimental CV-QKD systems still operate with Gaussian modulated coherent states [2,17,39]. Nevertheless, one cannot fail to note the incipient growth in the popularity of discrete modulation schemes [56,58,59].

*2.5. Quantum States*

Continuing the classification, in CV-QKD implementations, coherent or squeezed states can be used. They, in turn, can be single-mode or two-mode. An increase in the number of modes is also possible, but the appropriate analysis due to the structure of the states themselves (for example, when using phase-modulated attenuated coherent states) is often reduced to considering the single-mode case [51,61–63], so it does not make much practical sense to single it out separately.

First CV-QKD protocols were proposed in 1999 in [64]. These protocols utilized coherent and two-mode squeezed states with discrete modulation and homodyne detection.

In the coherent state protocol, pulse modulation is used to encode information using two modulators, amplitude and phase, and then one of the signal quadratures is detected. If the session is recognized as secure, the measurement results for the second quadrature are used for subsequent key generation. In the protocol operating with two squeezed states generated using two different squeezed light sources, an entangled state is considered. After homodyne detection of the modes of the entangled state, it becomes possible to extract information about one of the initial states.

Later, CV-QKD protocols were proposed based on squeezed states with Gaussian modulation, where the states were modulated by quadratures according to a two-dimensional Gaussian distribution [34]. Initially, the protocol was described in the context of homodyne detection, but later heterodyne detection was also taken into account [65]. The CV-QKD protocol proposed in 2002 with Gaussian modulation of coherent states GG02 can now be considered the most well-known and widespread [11,66].

Discussing coherent states further, thermal states should also be mentioned separately. In essence, they are being noisy coherent states themselves and require equivalent mathematical description. Currently, research is underway in the field of CV-QKD using such states. The security analysis of CV-QKD protocols using thermal Gaussian states in the case of collective Gaussian attacks was carried out, for example, in the work [67], for cases of direct and reverse reconciliation with homodyne and heterodyne detection methods. The authors have shown that in the case of direct reconciliation with homodyning detection, it is possible to increase the key rate when taking into account the Alice's trusted noise, even with large values of the modulation variance of the initial thermal states. In addition, the authors determined the upper bound of entanglement stability in the case of CV-QKD for different wavelength values.

Regarding practical implementation of CV-QKD systems, it should be noted that CV-QKD protocols based on squeezed states may be superior to protocols based on coherent states. However, the assumption of infinite squeezing [65] should be used, which cannot be implemented experimentally. In addition, the generation of squeezed states is a much more complex task, than the generation of coherent ones (in fact, they are states of attenuated laser radiation), which becomes the most problematic part of the implementation of the CV-QKD protocol using squeezed states in practice [68].

*2.6. Coherent Detection Schemes*

Coherent detection is based on mixing the source signal with the reference field of an external radiation source (LO), detecting mixed optical fields on a photosensitive site, and subsequent subtraction of signals on an electrical circuit. Homodyne and heterodyne methods are varieties of coherent detection, widely used in modern fiber-optical communication systems.

The reason to switch to coherent detection systems is their possibility to use various types of multilevel modulation. In the case of QKD systems, coherent detection makes it possible to register a weak signal without using single-photon detectors, which is the main advantage over DV-QKD systems.

2.6.1. Homodyne Detection

Homodyne detection is based on the process of the two-wave interference at the sensitive site of the photodetector.

As for the homodyne detection method, the frequency of a powerful LO is equal to the frequency of the signal $\omega_{LO} = \omega_S$. The general principle can be described by the equations of two-beam interference. Let the LO and signal fields depend on the coordinate $\vec{r}$ and the time $t$ as:

$$E_{LO}(\vec{r}, t) = A_{LO}(\vec{r}) \cos(\omega_{LO} t + \varphi_{LO}), \tag{1}$$

$$E_S(\vec{r}, t) = A_S(\vec{r}) \cos(\omega_S t + \varphi_S), \tag{2}$$

where $A_{LO}(\vec{r})$, $A_S$ are amplitudes of LO and signal, $\omega_{LO}, \omega_S$ are frequencies of LO and signal, $\varphi_{LO}, \varphi_S$ are phases of LO and signal.

Mixing is carried out on a 50/50 beam splitter. The interference result on each of the output arms of the beam splitter can be recorded either in the form of resultant fields or in the form of corresponding intensities:

$$I_1(\vec{r}, t) = A_{LO}^2(\vec{r}) + A_S^2(\vec{r}) + 2A_{LO}(\vec{r})A_S(\vec{r}) \cos(\varphi_{LO} - \varphi_S), \tag{3}$$

$$I_2(\vec{r}, t) = A_{LO}^2(\vec{r}) + A_S^2(\vec{r}) - 2A_{LO}(\vec{r})A_S(\vec{r}) \cos(\varphi_{LO} - \varphi_S). \tag{4}$$

The interaction of the total field with the sensitive area of the photodetector leads to the appearance of an output current proportional to the intensity. The occurrence of photoelectrons leads to the corresponding occurrence of current pulses in the recording circuit. The useful component of the signal can be easily separated from the so-called noise components in the subtracting electrical circuit. Therefore, for the difference current of the balanced detector, one can write:

$$i = 4S_{eff}A_{LO}A_S \cos(\varphi_{LO} - \varphi_S), \tag{5}$$

where $S_{eff}$ is a generalized constant containing the efficiency of a photosensitive material.

Depending on the argument of the harmonic function in Equation (5), one or another constant signal level will be observed. In a typical case for CV-QKD protocols with discrete modulation (for example, [61]), depending on the further phase selection by the receiver, either positive (phase match) or negative (phase mismatch) levels of the resulting signal will be observed. If the bases do not match, the resulting signal is uniformly distributed relative to zero. It is clear from Equation (5), that LO plays the role of an amplifier, since the amplitude of the useful signal is directly proportional to the amplitude of LO, and by increasing it, one can achieve the required level of the final signal. However, the restriction on it is imposed by the fact that noise is increasing together with power.

Since the choice of the LO phase determines the detectable quadrature of the field, this method can register only one quadrature of the boson field. In QKD schemes [52,61,64,68], using homodyne detection, one or another LO phase is randomly selected using a random number generator.

### 2.6.2. Heterodyne Detection

This method is an extension of a classical homodyne scheme with the opportunity to detect two quadratures of the field simultaneously in a single measurement. The scheme includes a 90-degree optical hybrid, where the signal is mixed in parallel with the LO. Its phase in one of the arms is shifted by $\pi/2$.

Thus, Bob does not need to choose a random phase for each state, that is, register one quadrature per measurement. In this case, it is necessary to take into account the additional losses, (3 dB), which appears due to the use of a 90-degree optical hybrid. In existing CV-QKD systems, this method is quite common [34,60,66,68].

This method is called "heterodyning" in a number of works, which contradicts the classical description of heterodyning [18] since the LO frequency coincides with the frequency of the signal. However, the name is justified by the fact that, as a result of detection, Bob has two signal quadratures. However, sometimes, the term "double homodyning" occurs [2].

### 2.6.3. Heterodyne Detection with Transfer to Difference Frequency

Classical laser heterodyning is based on the nonlinearity of a photodetector in relation to the radiation field [18]. If the sum of two harmonic signals undergoes a nonlinear transformation (in particular, quadratic), then harmonics with both total and difference frequencies appear as a result. Detecting an optical signal is nothing more than a quadratic transformation of the radiation field. Therefore, it is natural to expect that when two optical signals with different frequencies are detected simultaneously, an electrical signal at a difference frequency will appear at the output of the photodetector. CV-QKD systems using the heterodyne method with frequency transfer are rare [69,70], as the method from the previous section (Section 2.6.2) turns out to be more practically convenient [2].

With a heterodyne detection method with a transfer to a difference frequency, the LO frequency does not coincide with the signal frequency $\omega_{LO} \neq \omega_S$. Let $\omega_{LO} - \omega_S = \omega_-$.

When the fields are added together, components at difference frequencies will occur in this case, in contrast to the case of homodyne detection. Here, assuming that photodetectors do not perceive high (total) optical frequencies, but perceive a difference radio frequency, expressions for currents take the following forms:

$$i_1 = \frac{1}{2}S_{\text{eff}}(A_{LO}^2 + A_S^2 + 2A_{LO}A_S \cos(\omega_- t + \varphi_{LO} - \varphi_S)), \tag{6}$$

$$i_2 = \frac{1}{2}S_{\text{eff}}(A_{LO}^2 + A_S^2 - 2A_{LO}A_S \cos(\omega_- t + \varphi_{LO} - \varphi_S)). \tag{7}$$

Then, the difference current is:

$$i = 2S_{\text{eff}}A_{LO}A_S \cos(\omega_- t + \varphi_{LO} - \varphi_S). \tag{8}$$

At the same time, the received signal stores information about the amplitude and phase of the original signal, which can be obtained from the harmonic function. This approach makes it possible to detect two signal quadratures simultaneously, similar to double homodyning, since now Bob does not need to set any phase of the LO.

### 2.6.4. Coherent Detection and Protocol Security

Considering the issue of justifying the choice of a coherent detection method concerning a formal theoretical description of the protocol, in particular, in the context of justifying its durability, the requirement of symmetrization is a necessity for a security proof. Thus, it is necessary to consider the case where Alice and Bob work with both quadratures, which corresponds to the case of heterodyning. The implementation of the protocol symmetrization is carried out to simplify further theoretical analyses [31,71,72].

In the BB84 protocol paradigm, the standard symmetrization procedure is to implement a uniform permutation to $n$ states on both sides (both Alice and Bob)—they apply the same permutation to their states.

Similarly, in the case of CV-QKD, the same procedure can be performed, namely, the permutation of $n$ bosonic modes. However, for some protocols, a wider set of symmetries can be used—Alice and Bob can choose the same random unitary matrix from the group $U(n)$. Thus, the chosen unitary matrix will describe a passive linear transformation through the transformation of the annihilation operators $a \to Ua, b \to U^*b$, where $a$ and $b$ correspond to vectors of length $n$, each component of which is a mode destruction operator. If we apply such a transformation to the state $|\Phi\rangle^{\otimes n}$, where $|\Phi\rangle$ is a two-mode compressed vacuum state, it will remain invariant.

It should be noted that although the symmetrization procedure can be carried out with quantum states in a quantum channel directly, the complexity of its experimental implementation is unreasonably high. This leads to the need to work with directly measurable quantities; that is, the symmetrization must commute with the real measurement (with complex amplitudes of coherent states). Such a scenario corresponds to the case of heterodyne detection and cannot be implemented for homodyning.

### 2.7. Types of LO

Various approaches to CV-QKD protocols' security proofs use the standard assumption that Eve does not have access to the LO. However, in practice, Eve may have access to it, thus, opening up additional opportunities for attacks. One of the latter is discussed below as an example.

In practical CV-QKD systems, LO radiation is used as a reference, coherent with a weak quantum signal and, therefore, allows us to measure the quadrature of a weak field. However, it can also be used to generate a trigger signal needed to perform measurements. The LO signal, for example, can be appropriately modified by Eve, so that a trigger signal generated by the clock circuit also changes [73]. As a result, if legitimate users utilize a previously calibrated ratio to estimate shot noise based on the measured LO power, they will use its incorrect value in the event of a trigger signal delay. Thus, Eve can vary the value of shot noise by using an attack of the type "intercept-resend":

- Eve introduces an attenuator featuring an attenuation coefficient $0 \leq \alpha \leq 1$ into the channel for a fraction $0 \leq \nu \leq 1$ of LO pulses to change the shape of the pulses themselves. The trigger is delayed by $\delta$.
- Eve introduces a beam splitter featuring a transmission coefficient $0 \leq \mu \leq 1$ and realizes an attack as a partial "intercept-resend" [74]. Thus, the excess noise of the system is given by:

$$\xi_a = \xi + 2\mu N_0, \tag{9}$$

where $N_0$ is a shot noise and $\xi$ is an evaluation of excess noise without an attack.

When Eve increases the proportion of $\mu$ signal pulses for which partial intercept-resend attack is performed, then more noise is introduced. Obviously, the case when $\mu = 1$ describes a common version of the "intercept-resend".

The fraction $\nu$ of LO pulses attenuated by Eve and the attenuation coefficient $\alpha$ are the two variable parameters that play the same role to scale the variance of the measurements made by the receiver, making the value of the shot noise estimate unchanged. Thus, the presence of Eve in the channel is masked.

LO can be generated both in the Alice module (transmitted LO (TLO)) and in the Bob module (local LO (LLO)). In the first case, Alice generates LO and transmits it to Bob via a quantum channel, along with weak signal radiation. In this implementation, the reference signal is initially coherent to a weak quantum signal, but its intensity decreases according to the losses in the channel. In the case of LLO, the task of restoring coherence is set, the solution of which requires the implementation of additional practical solutions.

In practice, CV-QKD schemes with LLO feature relatively large excess noise since a signal pulse and the LO are generated from two independent lasers [75,76]. In contrast to the TLO scheme, the LLO scheme is more challenging in terms of establishing the phase reference. The reason for it is the fact that the relative phase varies with time due to the phase drift of the two lasers employed. Thus, each of the weak signal pulses is generally accompanied by a strong phase reference pulse, whose relative phase is used to estimate the relative phase of the signal pulse [77]. The frequency difference between Alice's source and Bob's LO should also be reasonably stable. Regarding integration into the existing telecommunication infrastructure, the very presence of additional independent laser sources can be an expensive solution.

LO sent via an optical channel is easier to implement but, in this case, the protection against side-channel attacks must be possible. The countermeasure for the attack mentioned consists of a real-time measurement of shot noise. Taking into account the linear dependence of the detector noise dispersion on the output power, it is possible to measure the LO power. An example of measurement is the application of strong attenuation on the signal path in Alice's module to a random set of pulses, using, for example, an optical switch or an amplitude modulator. Alternatively, additional homodyne detection systems can be utilized in the user modules. We also note the possibility of developing countermeasures using neural network algorithms [78,79]. It should also be noted that LLO does not guarantee protection against all relevant side-channel attacks, such as the polarization attack considered in the work [80].

### 2.8. Reconciliation Protocols

A separate important stage for CV-QKD is the reconciliation procedure. This process is determined by a separate general parameter, the reconciliation efficiency $\beta$:

$$\beta = \frac{H(Q(\mathbf{B})) - R^{\text{Source}}}{I(B;\,A)}, \tag{10}$$

$$R^{\text{Source}} \geq \sum_{i=0}^{m-1} H\left(B^i | \mathbf{A}, B^{i-1}, \ldots, B^0\right), \tag{11}$$

where $I$ is the mutual information, $H$ is Shannon entropy, $\mathbf{A}$, $\mathbf{B}$ are Alice and Bob sequences, $R^{\text{Source}}$ is a source coding rate, $Q$ is a digitizing function of an $m$-bit digital converter.

The efficiency of the reconciliation procedure directly affects the secure key generation rate and, as a result, determines the maximum key distribution distance in the practical implementations of CV-QKD systems.

One-way information reconciliation, when one party sends information about its key to another party, can be performed in two different ways: either Bob corrects his bits according to the Alice's data (direct reconciliation), or Alice corrects her bits according to Bob's data (reverse reconciliation) [24]. In the case of direct reconciliation with losses in the quantum channel of 3 dB, Eve potentially has more information about what Alice has prepared [2] than Bob, so the secure key cannot be generated. This loss limit can be overcome post-selection (deleting data with a given noise threshold [81]) or reverse reconciliation when Alice corrects her bits according to the receiver's data [82]. In the latter scenario, Alice's data is primary, and since Alice's information about the measurement results is always greater than Eve's information, the secure key generation rate can remain positive at higher losses. It should be noted that direct reconciliation is inferior to the reverse even with post-selection [2,37,52]. Thus, reverse reconciliation is more optimal.

Various reconciliation schemes have been proposed for CV-QKD with Gaussian modulation. The two main schemes are slice reconciliation [83] and multidimensional reconciliation [84]. Both schemes can use, for example, low-density parity-check codes or polar codes to correct errors [85].

The slice reconciliation method is more suitable for short distances and, theoretically, can allow for extracting several bits of information from a single pulse [83,86]. The essence of this method is to discretize potentially continuous values with subsequent processing.

In turn, the multidimensional reconciliation provides an effective way to correct Gaussian continuous variables with a low signal-to-noise ratio. The basic idea is to transform a Gaussian quantum channel into a virtual binary input additive white Gaussian noise (BI-AWGN) channel.

To present the method of multidimensional reverse reconciliation (Alice coordinates her data according to Bob's), further consideration will consider the example of eight-dimensional reconciliation [84]. As a result of the sequential completion of the generation, sending, detection, and sifting stages, Alice and Bob become joint owners of correlated Gaussian sequences. Let the vectors $X = (x_1, x_2, \ldots, x_8)^T$ and $Y = (y_1, y_2, \ldots, y_8)^T$ refer to Alice and Bob, respectively. Then, $X = Y + Z'$, where $Z'(0, \sigma'^2)$, and $\sigma'^2$ is a multidimensional variance of Gaussian noise. Then, the users normalize the Gaussian values: $x = X/\|X\|, y = Y/\|Y\|$, where the norm is $\|X\| = \sqrt{\langle X, X \rangle} = \sqrt{\sum_{i=1}^{8} x_i^2}$, expressions for $Y$ are similar to the replacement of notation. Therefore, the vectors $x$ and $y$ are uniformly distributed on the unit sphere $S^7$ over $S^8$. Bob generates a random sequence $(b_1, b_2, \ldots, b_8)$, and then it is displayed on a single sphere as:

$$u = \left( \frac{(-1)_{b_1}}{\sqrt{8}}, \frac{(-1)_{b_2}}{\sqrt{8}}, \ldots, \frac{(-1)_{b_8}}{\sqrt{8}} \right). \tag{12}$$

Then, Bob calculates the mapping $M(y, u)$, for which $M(y, u)x = u$ is executed. This operation is performed according to the rule:

$$M(y, u) = \sum_{i=1}^{8} \alpha_i(y, u) A_i, \tag{13}$$

where $\alpha(y, u) = (\alpha_1, \alpha_2, \ldots, \alpha_8)^T$ is the coordinate of the vector $u$ in the orthonormal basis $\{A_j y\}$.

Therefore, $\alpha(y, u) = (A_1 y, A_2 y, \ldots, A_8 y)^T u$, whereas $A_i$ is an orthogonal matrix. The latter in the context of the case under consideration may have the form:

$$A_1 = I_8, \tag{14}$$

$$A_2 = \{K_0, K_2\}, \tag{15}$$

$$A_4 = \{K_{00}, K_{32}, K_{20}, K_{12}\}, \tag{16}$$

$$A_8 = \{K_{000}, K_{332}, K_{320}, K_{312}, K_{200}, K_{102}, K_{123}, K_{121}\}, \tag{17}$$

$$K_0 = \begin{pmatrix} 1 & 0 \\ 0 & 1 \end{pmatrix}, \tag{18}$$

$$K_1 = \begin{pmatrix} 0 & 1 \\ 1 & 0 \end{pmatrix}, \tag{19}$$

$$K_2 = \begin{pmatrix} 0 & -1 \\ 1 & 0 \end{pmatrix}, \tag{20}$$

$$K_3 = \begin{pmatrix} 1 & 0 \\ 0 & -1 \end{pmatrix}. \tag{21}$$

Next, through an open classical channel, Bob sends $M(y, u)$ to Alice, while Alice calculates the value of $v = M(y, u)x$. Thus, a virtual Gaussian channel is organized with a binary input $u$ and a continuous output $v$. After the generation, sending, receiving, and sifting stages, Alice and Bob share the correlated Gaussian sequences. It can also be noted that in the case of work in the field of small values of the signal-to-noise ratio (i.e., below 0.5), the multidimensional reconciliation method achieves better results than the slice

reconciliation method. In the context of the work under consideration, it is the cross-section reconciliation scheme that should be considered optimal. With a small target distance, the use of this method obtains sufficient coordination efficiency, while having a significant advantage in the form of the relative simplicity of its implementation [15].

### 3. Conclusions

Summing up the work, one can present the optimal CV-QKD protocol, i.e., the final choice of the appropriate characteristic for each of the criteria stated above and discussed in detail:

- General scheme of the protocol—PM scenario;
- Quantum channel implementation—fiber-optical network;
- Channel configuration—one-way scheme;
- Type of modulation—Gaussian modulation;
- Signal states—single-mode coherent states;
- Coherent detection scheme—heterodyning;
- LO implementation—on Alice's side;
- Reconciliation protocol—reverse reconciliation.

Note, that the choice of exact solutions for the CV-QKD protocol is based on convenience of integration into existing network telecommunication infrastructures.

The study of CV-QKD protocols was conducted to increase the secure key generation rate exclusively in the context of proven security against collective attacks for finite-length keys. This work can be regarded as a preliminary stage for the next step—the creation of an experimental sample of the CV-QKD system operating in accordance with the selected protocol.

**Author Contributions:** Conceptualization, R.G.; methodology, R.G. and I.V.; investigation, R.G., I.V. and D.K. (Daniil Kirichenko); writing—original draft preparation, R.G., I.V. and D.K. (Daniil Kirichenko); writing—review and editing, R.G., I.V., I.F., I.A., V.C., S.S., B.N., B.P. and D.K. (Daria Kargina); supervision, E.S. and V.E.; project administration, E.S. All authors have read and agreed to the published version of the manuscript.

**Funding:** This research was funded by JSCo Russian Railways.

**Institutional Review Board Statement:** Not applicable.

**Informed Consent Statement:** Not applicable.

**Acknowledgments:** The work was completed by the Leading Research Center "National Center for Quantum Internet" of ITMO University by order of JSCo Russian Railways.

**Conflicts of Interest:** The authors declare no conflict of interest.

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
