# Peer review of "The Rationale for the Optimal Continuous-Variable Quantum Key Distribution Protocol"

_optics, doi:10.3390/opt3040030_

Round 1

Reviewer 1 Report

 Report on: 

Analytical review of the current state of continuous-variable quantum key distribution: The rationale for the optimal protocol 

The paper presents a classification scheme to establish an optimal protocol for CV-QKD. To achieve this objective, the authors revise two alternatives of each of the criteria employed in the experimental implementation of CV-QKD and analyse them in order to decide which is better. This is an interesting paper that could help in the development of CV-QKD. However, to be accepted for publication the paper must improve the following points: 

1) The abstract section is quite short, it must include the tools used to decide what alternative produces the optimal protocol. 

2) It has some typos, for example a) Although it is well known, the acronym QKD must be define before its first use. b) the word “key” is written twice in the second row of the introduction section, c) What the authors calls “entropy uncertainty relations” is known as entropic uncertainty relations, d) In the paragraph under the number 324, on page 9, it seems to be a typo in the definition of ? and y, it seems that the authors wants to write ?=?‖?|| and ?=?‖?||, this must be clarified. 

3) Maybe, the most important point to clarify to improve the paper is to establish a method to decide which alternative is better to implement of each of the Classification in the list given on Table 1. The alternatives on the classifications: i) type of modulation, ii) Coherent detection schemes, and iii) reconciliation protocols have enough explanation to elucidate which one of its alternatives is better than the other. 

However, the alternatives on the classifications: i) general scheme of a protocol, ii) quantum channel implementation, iii) channel configuration, iv) signal states, v) LO implementation, do not have a clear explanation of why the chose alternative is better than the other; for example, it is not clearly expressed why the prepare-and-measure alternative is better than the entangled based. 

4) The Conclusion section must suggest ideas for further research and must assess the impact of the paper. 

I think that these clarifications will improve the manuscript. I would recommend publication once the authors have considered the above suggestions 

Reviewer 2 Report

The author gives an overview of the existing CV-QKD researches and gives the conclusion of the optimal technology. However, I cannot recommend it for publication based on the following questions:

Firstly, the content of combing in CV-QKD field is not comprehensive. The author just gives a very general classification, and a lot of important technologies and protocols are lacked. For example, the CV-MDI QKD protocol, the research on practical security of CV-QKD systems, etc. As a very important class of CV-QKD protocol, CV-MDI QKD protocol is one of the important research contents in the field of CV-QKD, and this kind of protocols does not belong to the one-way scheme or two-way scheme in the author's current classification standard. At the same time, the measurement method used by this kind of protocols is CV Bell-state measurement, which also does not belong to any of the current author's measurement methods. In the existing classification standards of the author, how to classify CV-MDI QKD and how to classify CV Bell-state measurement are questions worth answering. At the same time, the practical security and theoretical security are also the standards to consider the security of CV-QKD, which are also not involved in the article. Due to the gap between actual devices and theoretical models, the existing CV-QKD system may have security vulnerabilities to be broken, so the attacks against these vulnerabilities and the defense methods against these attacks are also very important research contents, which are only slightly involved in Section 2.6. However, the attacks do not only utilize LO, but practical detector, practical source as well. These are significant and necessary to comb.

Secondly, as a review manuscript, the analysis of the article is not deep enough and is not representative. In other words, the comparison of various options in the manuscript is almost based on existing conclusions, and the author themselves has few opinions on these technologies and methods. In each section, the author only gives the advantages and disadvantages of various technical means of each part, which have been summarized and recognized. The author does not give his own principles and standards on how to choose each technical means in a specific scene.

Thirdly, the manuscript is not frontier enough, for it not fully express the development context of this field, recent research progress, problems to be solved and bottlenecks to be faced. For example, what is the motivation of each proposed protocol? What is the maximum achieved secret key rate and distance in the experiment at present? What is the maximum achieved reconciliation efficiency? What are the protocols that have been implemented in the experiment at present? What are the bottlenecks in the actual experiment of the protocols that have not been implemented? These questions are not answered in the manuscript, so it is not conducive for readers to grasp the development status of this field.

Fourthly, the conclusion of the manuscript is partial and arbitrary, thus is misleading. The conclusion about the best choice of each part of CV-QKD lacks the actual scene and more reasons. Specifically, the author only gives their respective advantages and disadvantages, and does not give the principles and standards of selection. In fact, the different choices of each part have their specific application scenarios and unique advantages, and the conclusion which is the best without a given application scenario cannot be given. For example, in practical experiments, perfect Gaussian modulation cannot be achieved, so discrete modulation may be a better choice at this time. In the case of difficult optical fiber deployment, free space CV-QKD may be a better choice. When the system requires higher security, the “locally” local oscillator scheme (LO in Bob’s side) may be a better choice, and so on. Thus, these conclusions are not valid in some specific scenarios. In other word, these conclusions are not accurate. Finally, this article seems to be prepared in a hurry without paying attention to the details, so some descriptions are confusing and some equations are wrong. For example, the symplectic matrix of a beam splitter is wrong, the normalization equation is wrong, etc. In addition, the author claims that “CV-QKD being the case, Alice generates a two-mode squeezed vacuum state and measures both quadratures of one of its modes” in line 55-56. However, this sentence is ambiguous. Alice generates a two-mode squeezed vacuum state, measures both quadratures of one of its modes, then another mode is projected onto a coherent state. In other words, this operation only can generate coherent states. This is only a case in CV-QKD, not all case. If Alice randomly measures one of the quadratures of a mode, she will project another mode onto a squeezed state. Thus, the description here is confusing and may mislead the readers. In addition, the authors claim that “Coherent detection is based on mixing the source signal with the reference field of an external radiation source — LO, detecting mixed optical fields on a photosensitive site and subsequent subtraction of signals on an electrical circuit” and CV-QKD is such protocols based on coherent detection. Based on the reality that CV-MDI QKD protocol is one of CV-QKD protocols, the used detection method CV Bell-state measurement is coherent detection too. However, a very important step in CV Bell-state measurement before detection is to use a 50:50 beam splitter to interfere with two quantum level signals, which is different from ordinary coherent detection. Therefore, a more detailed classification standard or explanation is required here. The current version may be misleading.

In a word, I think the depth, scope, accuracy and foresight of this manuscript are not enough, so it cannot be published.

Round 2

Reviewer 2 Report

The authors give responses to most of the questions and revised the article accordingly. However, there are still some problems with the current version. Firstly, the matrix of the beam splitter in Eq. (3) is still wrong in my opinion. Secondly, I think more explanations are needed to reach the final conclusion. For example, according to the final conclusion, the authors think that it is easier to integrate the scheme that local oscillator is generated in Alice’s side with the current telecommunication network. I think this conclusion needs more explanation. In the manuscript, the authors claim that the local oscillator sent via an optical channel is easier to implement, but there are also side channel attacks to be considered. In order to solve this problem, additional practical solutions also need to be done. Why does the authors think that the safer “locally” local oscillator scheme is more difficult to integrate into the current network than the scheme in which local oscillator is generated in Alice’s side?
